

# Measurements of NO and NO₂ exchange between the atmosphere and *Quercus agrifolia*

Erin R. Delaria[1], Megan Vieira,[1] Julie Cremieux[2], Ronald. C. Cohen[1,3]

[1]Department of Chemistry, University of California, Berkeley, 94720, USA
[2]Department of Chemistry, Université Pierre et Marie Curie, Paris, France
[3]Department of Earth and Planetary Science, University of California, Berkeley, 94720, USA

*Correspondence to*: R. C. Cohen (rccohen@berkeley.edu)

**Abstract.** $NO_2$ foliar deposition through the stomata of leaves has been identified as a significant sink of $NO_x$ within a forest canopy. In this study, we investigated $NO_2$ and NO exchange between the atmosphere and the leaves of the native California
oak tree *Quercus agrifolia* using a branch enclosure system. $NO_2$ detection was performed with laser-induced fluorescence (LIF), which excludes biases from other reactive nitrogen compounds and has a low detection limit of 5–50 ppt. We performed both light and dark experiments with concentrations between 0.5–10 ppb $NO_2$ and NO under constant ambient conditions. Deposition velocities for $NO_2$ during light and dark experiments were $0.123 \pm 0.007$ cm s$^{-1}$ and $0.015 \pm 0.001$ cm s$^{-1}$, respectively. Much slower deposition was seen for NO, with deposition velocities of $0.012 \pm 0.002$ cm s$^{-1}$ and $0.005 \pm$
$0.002$ cm s$^{-1}$ measured during light and dark experiments, respectively. This corresponded to a summed resistance of the stomata and mesophyll of $6.9 \pm 0.9$ s cm$^{-1}$ for $NO_2$ and $140 \pm 40$ s cm$^{-1}$ for NO. No significant compensation point was detected for $NO_2$ uptake, but compensation points ranging from 0.74–3.8 ppb were observed for NO. $NO_2$ and NO deposition velocities reported here are comparable both with previous leaf-level chamber studies and inferences from canopy-level field measurements. In parallel with these laboratory experiments, we have constructed a detailed 1-D
atmospheric model to assess the contribution of leaf-level $NO_x$ deposition to the total $NO_x$ loss and $NO_x$ canopy fluxes. Using the leaf uptake rates measured in the laboratory, these modeling studies suggest loss of $NO_x$ to deposition in a California oak woodland competes with the pathways of $HNO_3$ and $RONO_2$ formation, with deposition making up 3–22% of the total $NO_x$ loss. Additionally, foliar uptake of $NO_x$ at these rates could account for ~15–30% canopy reduction of soil $NO_x$ emissions.

## 1 Introduction

Nitrogen oxides ($NO_x \equiv NO + NO_2$) are a group of highly reactive trace gases that control the oxidative capacity of the atmosphere, regulating the amounts of ozone, hydroxyl radicals, volatile organic compounds, and other key atmospheric species (Crutzen, 1979). $NO_x$ is also directly toxic in high concentrations, plays a significant role in tropospheric ozone production, and serves as a source of $NO_3^-$, a key nutrient for ecosystems and significant component of acid rain. $NO_x$ is
primarily emitted as nitric oxide (NO) through fossil fuel combustion, biomass burning, lightning and microbial activity in



soils (Seinfeld and Pandis, 2006). NO is rapidly oxidized to nitrogen dioxide ($NO_2$) through reactions with ozone and peroxy radicals, and $NO_2$ subsequently photolyzes to reform NO. The interconversion of NO and $NO_2$ reaches steady-state within a few minutes during the daytime (Crutzen, 1979). The effects of $NO_x$ on urban chemistry, where anthropogenic emissions dominate the $NO_x$ source, have been extensively studied. However, the processes affecting $NO_x$ in forested and agricultural

regions are less well-understood.

In forests and agricultural lands, the major source of $NO_x$ is NO emitted as a by-product of microbial denitrification and nitrification (Mckenney et al., 1982; Caranto and Lancaster, 2017). Deposition of $NO_2$ to plant canopies is thought to be a significant sink of $NO_x$ in forests, substantially reducing the contribution of soil-emitted $NO_x$ to the atmospheric $NO_x$ budget. Jacob and Wofsy (1990) observed low $NO_x$ above the canopy over the Amazon forest during the wet season. Using a

1-D chemical and transport model constrained by observed $NO_x$ and ozone, they concluded that a substantial fraction of soil-$NO_x$ must be absorbed by the canopy. Extrapolation of these ideas to forests with different leaf area indices suggest that 20–50% of the global fraction of soil-emitted $NO_x$ is lost to vegetation (Yienger and Levy, 1995; Lerdau et al., 2000). Using the framework of Jacob and Wofsy (1990), and Yienger and Levy (1995), global atmospheric models have been tuned to describe observed atmospheric $NO_x$ concentrations and tropospheric ozone production using a canopy reduction factor

(CRF), an adjustable parameter which accounts for the difference between soil NO emissions and the amount of $NO_x$ ventilated through the canopy (Yienger and Levy, 1995; Ganzeveld et al., 2002a; Vinken et al., 2014). However, CRFs are implemented in an unphysical manner where they act only on soil $NO_x$ emissions and not on other $NO_x$ present in the plant canopy. An improved understanding is needed of the physical processes governing the foliar uptake of $NO_x$ at the ecosystem and leaf scales.

Many studies have also directly observed the leaf-level uptake of $NO_2$ (Neubert et al., 1993; Rondon and Granat, 1994; Hereid and Monson, 2001; Sparks et al., 2001; Teklemariam and Sparks, 2006; Pape et al., 2009; Chaparro-Suarez et al., 2011; Breuninger et al., 2013). Experiments investigating the mechanism of $NO_2$ uptake using the $^{15}N$ isotope as a tracer have demonstrated that atmospheric $NO_2$ can be absorbed through the stomata of plant leaves, converted to nitrate ($NO_3^-$) and nitrite ($NO_2^-$), and eventually assimilated into amino acids (Rogers et al., 1979; Okano and Totsuka, 1986; Nussbaum et

al., 1993; Weber et al., 1995; Yoneyame et al., 2003). The mechanism of $NO_2$ assimilation is diffusion into the stomata followed by dissolution into the aqueous phase and disproportionation to $NO_3^-$ and $NO_2^-$ in the apoplasm (Lee and Schwartz, 1981a, b). $NO_2$ can also be transformed to nitrate and nitrite through scavenging by antioxidants, most notably ascorbate (Ramge et al., 1993). The influence of ascorbate on foliar uptake was theoretically calculated by Ramge et al. (1993), and experimentally demonstrated by Teklemariam and Sparks. (2006). The enzyme nitrate reductase converts $NO_3^-$ to $NO_2^-$ in

the cytosol. $NO_2^-$ is then transported into the plastids where it is further reduced by the enzyme nitrite reductase to ammonium ($NH_4^+$), the product required for amino acid synthesis (Ammann et al., 1995; Tischner, 2000; Teklemariam and Sparks, 2006). Alternatively, $NO_2$ can deposit directly onto the leaf cuticles or a leaf-surface water film (Burkhardt and Eiden, 1994). However, foliar uptake of $NO_2$ has been demonstrated to be controlled primarily by the stomata, with deposition to the leaf surface representing only a small fraction of the total $NO_2$ flux (Thoene et al., 1991; Gessler et al.,





2000; Chaparro-Suarez et al., 2011). Strong correlations have been observed between $NO_2$ concentrations, stomatal conductances, and the $NO_2$ deposition flux, suggesting foliar uptake is mainly controlled by stomatal aperture and internal leaf resistances (Johansson, 1987; Thoene et al., 1991; Rondon et al., 1993; Meixner et al., 1997; Chaparro-Suarez et al., 2011; Breuninger et al., 2013).

Despite the large body of research that exists on the leaf-level deposition of $NO_2$ to vegetation, discrepancies still exist of $NO_2$ exchange rates and the role of mesophilic processes. Many laboratory experiments have failed to measure uptake rates necessary to describe the observed 20–50% reduction of soil-emitted $NO_x$ (Hanson and Lindberg, 1991; Breuninger et al., 2013). Another considerable controversy is the existence of a compensation point—a concentration below which leaves would instead act as a source of $NO_x$. Compensation points of 0.1–3.2 ppb $NO_x$ have been observed in a

number of laboratory chamber studies, suggesting trees instead may serve as a large source of $NO_x$ in forests (Johansson, 1987; Rondon et al., 1993; Hereid and Monson, 2001; Sparks et al., 2001; Teklemariam and Sparks, 2006). Emission of NO at these low $NO_x$ mixing ratios has also been detected in laboratory chamber studies (Wildt et al., 1997; Hereid and Monson, 2001). Observations of $NO_x$ canopy fluxes and atmospheric models conversely predict that trees are substantial sinks of $NO_x$ at concentrations as low as 0.1 ppb, typical of $NO_x$ mixing ratios in remote areas (Jacob and Wofsy, 1990). More recent

laboratory studies of leaf level deposition have also questioned the existence of a compensation point (Chaparro-Suarez et al., 2011; Breuninger et al., 2013).

Many laboratory investigations of $NO_x$ foliar exchange have not been performed with instruments sufficiently sensitive or specific to measure uptake of $NO_2$ at the low $NO_x$ concentrations relevant to forested environments. A commonly used technique for chamber observations of leaf-level $NO_2$ uptake is the indirect $NO_2$ measurement technique of

chemiluminescence detection of NO (Sparks et al., 2001; Teklemariam and Sparks, 2006; Pape et al., 2009; Chaparro-Suarez et al., 2011; Breuninger et al., 2012; Breuninger et al., 2013). This technique requires photolytic or catalytic conversion of $NO_2$ to NO, which is either limited by large detection limits greater than 100 ppt (Teklemariam and Sparks, 2006; Pape et al., 2009; Chaparro-Suarez et al., 2011; Breuninger et al., 2012; Breuninger et al., 2013), or may be subject to interferences from higher oxides of nitrogen (Sparks et al., 2001). Further, interferences from the chemiluminescence of alkene + ozone

reaction products have also been identified (Reed et al., 2016). These are of particular importance since alkenes make up a substantial fraction of biogenic VOC emissions (e.g. isoprene) (Kesselmeier et al., 2002; Lappalainen et al., 2009; Park et al., 2014; Romer et al., 2016). Even in laboratory settings, where interferences from higher oxides are not of concern, emissions of alkenes from the enclosed leaves may cause substantial interferences. New methods for studying the exchange at the leaf-level, are required to resolve existing discrepancies regarding the foliar uptake rate of $NO_2$ and the existence of a

compensation point.

To understand the leaf-level processes affecting ecosystem scale atmosphere-biosphere $NO_x$ exchange, we have conducted laboratory experiments measuring NO and $NO_2$ fluxes to the native California tree species *Quercus agrifolia* (Fig. 1) using a branch enclosure system and direct laser-induced fluorescence (LIF) detection of $NO_2$ (Fig. 2). With the LIF technique, we are able measure $NO_x$ exchange fluxes with high specificity and sensitivity. We investigated the existence of



an NO$_2$ and NO compensation point and the rate of NO$_x$ foliar uptake under controlled conditions. To our knowledge this is the first leaf-level uptake experiment that has been performed on a North American tree species.

## 2 Materials and methods

### 2.1 *Quercus agrifolia* samples

5   NO$_x$ uptake by *Quercus agrifolia* (Coastal Live Oak) was investigated in the laboratory. Three *Quercus agrifolia* individuals were purchased from a local native California plant nursery (Native Here Now Nursery), where the plants were grown from seeds and cuttings collected in Contra Costa County. The tree specimens were grown in a nutrient rich commercial soil mixture (a mixture of Orchard Potting Soil and EB Stone Cactus Mix) at the Jane Grey Research Greenhouse at the University of California, Berkeley. The trees were 2–3 years old when measurements were taken.

### 2.2 Laser-induced fluorescence detection

NO$_2$ was measured using Laser-Induced Fluorescence (LIF). A blue diode laser (Z-Laser ZM18H3,) centered at a wavelength of 405 nm was focused into each detection cell and made 20 passes in White multipass optical configuration (Fig. 2b)(Thornton et al., 2000; Fuchs et al., 2009). Upon absorption of a visible photon, NO$_2$ undergoes a transition from the $^2A_1$ ground to the $^2B_2$ excited electronic state. The excited NO$_2$ molecule, is either quenched by collision or emits a red-shifted photon as it relaxes back to ground state (e.g. Thornton et al., 2000). These emitted photons were detected using a red-sensitive photomultiplier tube (PMT) (Hamamatsu H7421-50). To minimize collisional quenching, each detection cell was maintained at a pressure of around 3 torr. Excitation at 405 nm was chosen because it is near the region of maximum absorption in the NO$_2$ spectrum, and is not subject to interferences from absorption by water vapor or O$_3$ (Matsumoto and Kajii, 2003).

20   Calibrations were performed every hour by diluting NO (4.97 ppm $\pm$ 5%, Praxair) and NO$_2$ standard gases (5.08 ppm $\pm$ 5%, Praxair) to 1–10 ppb in humidified (RH ~60%) zero air. The limit of detection (LOD) for the detection cells is described as:

$$LOD = \frac{S/N}{m}\sqrt{\frac{2b}{t}} \tag{1}$$

where m is the slope of the calibration curve constructed from standard dilutions, b is the PMT signal at 0 ppb NO or NO$_2$, S/N is the desired signal to noise ratio, and $t$ is the time of signal averaging. At a S/N of 2 and signal averaging over 5 min, the LOD for detection cells 1–4 was 15 ppt, 4 ppt, 10 ppt, and 30 ppt, respectively. NO$_2$ in the incoming and outgoing airstreams was measured simultaneously in the first two detection cells. In the second two detection cells, NO was quantitatively converted to NO$_2$ in the presence of excess ozone, allowing for detection of total NO$_x$ (Fig. 2a). Ozone was produced using an ozone generator (Jelight 600) and flow rates of ozone delivered were adjusted to achieve unity conversion of NO to NO$_2$.





Much of the previous work on leaf-level $NO_2$ uptake has been performed using indirect detection of $NO_2$, in which photolytic or catalytic conversion of $NO_2$ is followed by chemiluminescence measurement of NO (Sparks et al., 2001; Teklemariam and Sparks, 2006; Pape et al., 2009; Chaparro-Suarez et al., 2011; Breuninger et al., 2012; Breuninger et al., 2013). However, these techniques are either limited by much larger detection limits in the 100–500 ppt range, or are non-

specific in their conversion of $NO_2$ to NO. Our measurements demonstrate a much higher degree of certainty, due largely to a lower detection limit than comparable experiments with specific photolytic conversion and chemiluminescence measurement of $NO_2$ (Chaparro-Suarez et al., 2011; Breuninger et al., 2012; Breuninger et al., 2013). With the LIF detection of $NO_2$, we are able to sensitively measure exchange fluxes at low $NO_x$ mixing ratios relevant to forested environments.

## 2.3 Dynamic chamber system

The $NO_x$ flux measurements were performed with a dynamic branch enclosure system, consisting of a thin transparent double-walled Teflon film (FEP) bag (American Duraflim), which transmits 90% of photosynthetically activated radiation. The chamber was illuminated by an LED diode array of 430–475 nm and 620–670 nm lights (Apollo Horticulture). This light source was selected because it does not emit wavelengths below 420 nm, where $NO_2$ dissociates, preventing loss of $NO_2$ to photodissociation and resultant photochemistry. In order to ensure turbulent mixing and minimal aerodynamic and

boundary layer resistances, a Teflon-coated fan was installed inside the inner chamber (Meixner et al., 1997; Pape et al., 2009; Breuninger et al., 2013).

Before experiments with *Quercus agrifolia* individuals, the deposition to an empty chamber was measured and background subtracted from subsequent branch measurements. The measured loss of $NO_2$ to chamber walls was 5% of the $NO_2$ mixing ratio flowing into chamber. This corresponded to a maximum loss of 0.4 ppb at 8 ppb $NO_2$. Emission of less

than 0.05 ppb $NO_2$ from the Teflon walls was also observed when chamber lights were turned on with 0 ppb $NO_2$ flowing through the system. It is likely that the chamber walls buffer uptake of $NO_2$, but this is a minor effect, as the wall emission observed was a tiny fraction of the measured fluxes.

During measurements, the enclosed branch was exposed to known amounts of either $NO_2$ or NO mixed with zero air. The inner chamber had an inner diameter of 20 cm, a length of 40 cm, and a total volume of 13 L (American Durafilm

200A Teflon FEP). Flow rates into the inner chamber (Q) during experiments were typically 5 L min$^{-1}$, creating residence time in the chamber of 3 min. The outer chamber had an inner diameter of 30 cm and a length of 55 cm (American Durafilm 500C20 Teflon FEP). Zero air at a flow rate of 3 L min$^{-1}$ constantly fumigated the outer bag, serving as a buffer region to ensure the laboratory air, with high mixing ratios of $NO_x$, did not diffuse into the bag enclosing the branch.

The photosynthetic photon flux density (PPFD) was monitored outside the chamber with a LiCor quantum sensor

(LiCor LI-190SA). The flux density measured above the chamber was 1190 $\mu$mol m$^{-2}$ s$^{-1}$, approximately the PPF for Berkeley, California at noon during the month of October. This is well above the photon flux required to achieve maximal stomatal aperture for broadleaf evergreen trees (von Caemmerer and Farquhar, 1981; Chaparro-Suarez et al., 2011; Breuninger et al., 2013). We confirmed this assumption by covering the lights with a filter to reduce the intensity by 40%



and monitoring $CO_2$ and $H_2O$ exchange. No reduction in the exchange rates of these gases were observed. The relative humidity of air entering the chamber was maintained at 50–65% in all experiments by flowing zero air through a bubbler before mixing with $NO_x$. Measurements of $NO_x$ exchange fluxes occurred under a light/dark cycle with a photoperiod of 12 hours and a temperature of $26/22 \pm 2$ °C. No change in $NO_x$ uptake was observed when heating the chamber with the lights off or cooling the chamber with the lights on. We therefore expect no significant temperature effects caused by the 4°C difference in temperature between light and dark periods. We also observed a relative humidity increase in the delivered air of about 2% with the lights off, but do not expect this increase to produce any significant changes in $NO_x$ deposition or plant physiology (von Caemmerer and Farquhar, 1981; Chaparro-Suarez et al., 2011).

Exchange of $CO_2$ and $H_2O$ with the leaves were monitored with a LiCor-6262 $H_2O/CO_2$ analyzer operating in differential mode. Flows of 0.1 L min$^{-1}$ of air entering and exiting the chamber were diverted to the LiCor analyzer to measure the $CO_2$ assimilation and transpiration rates. To measure the $CO_2$ content and relative humidity of air delivered to the chamber, 0.5 L min$^{-1}$ of the humidified zero air/$NO_x$ mixture was diverted to a second external 1.5 L cuvette. The temperature and relative humidity of air entering the chamber were measured with a temperature and relative humidity module in the external cuvette (TE Connectivity HTM2500LF). The $CO_2$ mixing ratios in the external chamber were monitored with a Vaisala CarboCap GMP343 sensor.

## 2.4 $NO_x$ flux densities

The leaf-level exchange flux of NO or $NO_2$ ($F_{NO_x}$) was calculated according to Eq. 2:

$$F_{NO_x} = \frac{Q \cdot (C_0 - C_i)}{A} \tag{2}$$

where Q is the flow rate (m$^3$ s$^{-1}$), A is the enclosed leaf area (m$^2$), $C_0$ is the concentration leaving the chamber, and $C_i$ is the concentration entering the chamber (nmol m$^{-3}$). The calculated flux is related to a deposition velocity ($Vd_{NO_x}$) by Eq. 3:

$$F_{NO_x} = -Vd_{NO_x} \cdot (C_0 + C_{comp}) \tag{3}$$

where $C_{comp}$ is the compensation point, the concentration of $NO_2$ below which the tree would instead act as a source of $NO_x$.

The deposition velocities were calculated through weighted least squares regression of calculated fluxes and outlet $NO_x$ concentrations ($C_o$). The absolute value of the slope of the regression line was equal to the deposition velocity, with the x-intercept representing the compensation point concentration. The precision error in the $NO_x$ exchange flux ($\sigma_F$) was calculated through propagation of the error in the inlet ($\sigma_{C_i}$) and outlet ($\sigma_{C_o}$) concentrations (Eq. 4).

$$\sigma_F = \frac{Q}{A} \sqrt{\sigma_{C_i}^2 + \sigma_{C_o}^2} \tag{4}$$

$\sigma_{C_i}$ and $\sigma_{C_o}$ were estimated as the larger of the error in the calibration slopes and the standard deviation of the 5 min signal average. From observations in daily deviations of the flow rate and error in measured leaf area using the ImageJ software (Schneider et al., 2012), we estimate the error in $\frac{Q}{A}$ to be a maximum of 0.005 cm s$^{-1}$. This usually was only a minor contribution to the total error in the $NO_x$ exchange flux.



The calculated deposition velocity was used to find the total resistance to deposition, $R$, via Eq. 5.

$$Vd_{NO_x} = \frac{1}{R} \tag{5}$$

The total resistance is described by the canopy stomatal resistance model (Baldocchi et al., 1987) and defined in Eq. 6–7.

$$R = R_a + R_b + R_{leaf} \tag{6}$$

$$R_{leaf} = \left(\frac{1}{R_{cut}} + \frac{1}{R_{st}+R_m}\right)^{-1} \tag{7}$$

where $R_{leaf}$ is the total leaf resistance and $R_a$, $R_b$, $R_{cut}$, $R_{st}$, and $R_m$ are the aerodynamic, boundary layer, cuticular, stomatal, and mesophilic resistances, respectively. The aerodynamic resistance is characterized by the micrometeorology above a surface and is dependent upon the wind speed and turbulence of air flow. The boundary layer resistance describes the diffusion of a molecule through a shallow boundary of air above a surface and is dependent on microscopic surface

properties, diffusivity of the gas species, wind speed, and turbulence of air flow (Baldocchi et al., 1987). $R_{cut}$, $R_{st}$, and $R_m$ are the resistances associated with deposition to the leaf cuticles or through the stomata, and are dependent upon plant physiology.

The chamber fan, installed to create turbulent mixing, allowed for the assumption that $R_a$ was negligible (Pape et al., 2009; Breuninger et al., 2012). $R_b$ is chamber-specific, and has typically not been measured in previous chamber

experiments of $NO_2$ leaf-level deposition (Chaparro-Suarez et al., 2011; Breuninger et al., 2012; Breuninger et al., 2013). $R_b$ was experimentally measured in this study by placing a tray of activated carbon into the chamber (assumed to have zero surface resistance to deposition of $NO_2$), and calculating the deposition flux of $NO_2$. The leaf components to the total deposition resistance were determined through dark and light experiments. During dark experiments, the stomata were closed (confirmed with measurements of $CO_2$ and $H_2O$ exchange), and the deposition observed was assumed to be entirely

driven by deposition to the cuticles.

## 3 Results

### 3.1 Determination of the boundary resistance $R_b$

To estimate the chamber boundary layer resistance and test the assumption that $R_b \ll R_{leaf}$, a dish of activated carbon, which theoretically has zero chemical resistance to deposition of $NO_2$, was placed inside the chamber. The boundary layer

resistance was considered to be the only component of the total resistance to deposition. The deposition velocity of $NO_2$ to activated carbon was measured as $0.52 \pm 0.06$ cm s$^{-1}$, corresponding to a boundary layer resistance to $NO_2$ deposition of $1.94 \pm 0.02$ s cm$^{-1}$ (Fig. 3). This boundary resistance is approximately double what was measured by Pape et al. 2009—a reasonable difference given differences in chamber design (Fig. 2). The $R_b$ for $NO_2$ was scaled with the ratio of diffusivities of $NO_2$ and NO in air to obtain the resistance to deposition of NO of $2.59 \pm 0.03$ s cm$^{-1}$.



The boundary resistance was also estimated in an additional experiment (not shown) in which a de-ionized water-soaked Whatman No. 1 filter paper was placed inside the chamber and the evaporation of water vapor into the chamber filled with dry zero air was measured. The emission flux of water vapor from the filter paper was calculated in a similar manner to that of $NO_x$ deposition flux (Eq. 2). The conductance to water vapor was then calculated via:

$$\frac{Q \cdot (P_{H_2O})}{A} = g_w(P_{sat} - P_{H_2O}) \tag{8}$$

where $P_{H_2O}$ is the partial pressure of water vapor inside the chamber, $P_{sat}$ is the saturation vapor pressure at the temperature in the chamber, and $g_w$ is the conductance to water vapor. The measured conductance to water vapor was scaled with the ratio of diffusivities of $NO_2$ to water vapor ($D_{NO_2}/D_{H_2O}$) and inverted to find the $NO_2$ boundary layer resistance:

$$R_b = \frac{D_{H_2O}}{D_{NO_2}} \frac{1}{g_w} \tag{9}$$

The boundary resistance to $NO_2$ deposition by this method was found to be 2 s cm$^{-1}$, essentially identical to the measurement on the activated-carbon.

## 3.2 $NO_x$ deposition velocity and compensation point concentration

The deposition velocities and compensation points were respectively calculated as the slope x-axis intercept of the regression line between $NO_x$ exchange flux and chamber $NO_x$ concentrations (Fig. 4). The detection limit was a dominant source of error in the estimation of the NO exchange flux and compensation point. The large relative uncertainties in NO flux measurements were caused by the much slower deposition of NO compared with that of $NO_2$, inhibiting our ability to observe the very small changes between the NO concentration in the chamber and the incoming airstream (Fig. 4). Additional uncertainty in $NO_2$ flux measurements because of enhanced quenching of $NO_2$ by water vapor should be minimal, as calibrations and measurements were performed at equivalent relative humidities. However, transpiration of the enclosed leaves caused the absolute humidity within chamber to be enhanced by 0.3–0.5% relative to the incoming airstream. We expect this to result in a maximum error in calculated $NO_2$ mixing ratios of 1–1.75% (Thornton et al., 2000), resulting in maximum errors in the calculated fluxes and deposition velocities of 2% and 4%, respectively. This 4% error in the calculated deposition velocity during lights-on experiments is considerably less that the uncertainty of the linear fit (Fig. 4).

Correlation coefficients, deposition velocities, compensation points, and statistical testing of the compensation point for $NO_2$ and NO deposition are shown in Table 1 and Table 2, respectively, and were calculated according to Breuninger et al. (2013). For $NO_2$ experiments, only one dark and one light experiment with *Quercus agrifolia 1*, was found to have a statistically significant ($\alpha = 0.05$) non-zero intersection with the x-axis (Table 1). The range of $C_{comp}$ measured were -0.02–0.300 ppb $NO_2$, with probabilities of $C_{comp} = 0$ ranging from 10.3–91.6% (excluding the two *Quercus agrifolia 1* experiments) (Table 1). Conversely, all three *Quercus agrifolia* individuals during all dark and light NO deposition experiments demonstrated compensation points significantly above zero, ranging from 0.74–3.8 ppb NO. For all light and



dark experiments the average compensation point for was calculated as $0.84 \pm 0.32$ ppb NO and $2.4 \pm 1.1$ ppb NO, respectively (Table 2).

Student's t tests, (not shown), demonstrated that deposition velocities and compensation points measured during NO and $NO_2$ lights on and off experiments were not significantly different (to the $\alpha = 0.05$ confidence level) between different *Quercus agrifolia* individuals. Deposition velocities for $NO_2$ light experiments were between 0.08 and 0.18 cm s⁻¹, with a deposition of $0.123 \pm 0.007$ cm s⁻¹ calculated from the regression of all light experiments. Dark experiments resulted in deposition velocities between 0.013 and 0.022 cm s⁻¹, with a deposition velocity of $0.015 \pm 0.001$ cm s⁻¹ calculated from the regression of all dark experiments (Table 1). NO demonstrated much slower deposition, with deposition velocities from all light and dark experiments calculated as $0.012 \pm 0.002$ cm s⁻¹ and $0.005 \pm 0.002$ cm s⁻¹, respectively (Table 2). Despite the large compensation point measured for NO, the leaf emission fluxes of NO were a maximum of only 0.8 pmol m⁻² s⁻¹ at 0.1 ppb NO, approximately half of the deposition flux measured for $NO_2$ at 0.1 ppb, making *Quercus agrifolia* a large net sink of $NO_x$.

**3.3 Resistances to Leaf-level $NO_x$ deposition.**

The deposition velocity measured from linear regression of $NO_x$ exchange fluxes and $NO_x$ chamber concentrations is the inverse of the total resistance to deposition (Eq. 6), with $R_a$ assumed to be zero. The total resistance in the chamber is thus:

$$R = R_b + \left( \frac{1}{R_{cut}} + \frac{1}{R_{st}+R_m} \right)^{-1} \tag{10}$$

The leaf resistance to deposition can then be found by subtracting the boundary layer resistance from the total resistance. Total leaf resistances, $R_{leaf}$, were calculated using the boundary layer resistances for $NO_2$ and NO of $1.94 \pm 0.02$ s cm⁻¹ and $2.59 \pm 0.03$ s cm⁻¹, respectively. During the dark experiments, $R_{leaf}$ is equal to $R_{cut}$, and the deposition velocity measured was estimated as the inverse of the sum of the boundary and cuticular resistance. After calculation of $R_{cut}$ from dark experiments, the sum of the stomatal and mesophilic contributions to the total leaf resistance was determined. The boundary, cuticular, and summed stomatal and mesophilic resistances are shown in Table 3.

It must be noted that it is possible that the stomata were not entirely closed during dark experiments. Evidence exists that nocturnal stomatal conductance can be large enough to allow for significant transpiration (Dawson et al., 2007), and small (within the range of uncertainty observed for the LICOR-6262) emission of water vapor during dark experiments was measured. However, even if all the deposition during dark experiments was stomatal, this would cause only a 0.5 s cm⁻¹ reduction in the calculated $R_{st}$ for $NO_2$, less than the uncertainty from the error in the measured deposition velocity (~10% error). The cuticular resistances reported here during dark experiment are nonetheless atmospherically relevant to nighttime $NO_x$ deposition.





## 4 Discussion

### 4.1 NO$_x$ deposition velocities and compensation points

The strong linear dependence between NO$_2$ fluxes and NO$_2$ chamber concentrations that we observe is consistent with previous observations that NO$_2$ exchange is largely driven by NO$_2$ concentration differences between the atmosphere and

gaseous phase of the leaf (Rondon and Granat, 1994; Gessler et al., 2000; Hereid and Monson, 2001; Sparks et al., 2001; Teklemariam and Sparks, 2006; Pape et al., 2009; Chaparro-Suarez et al., 2011; Breuninger et al., 2012). Our measurements of NO$_2$ stomatal resistance parameters for *Quercus agrifolia* represents a stomatal deposition velocity (inverse of $R_{st} + R_m$) of $0.14 \pm 0.02$ cm s$^{-1}$. This value is similar to the range of 0.1–0.15 cm s$^{-1}$ that Chapparo-Suarez et al. (2011) found for two European oak tree species, *Quercus robur* and *Quercus ilex*. The deposition velocity measured here for *Quercus agrifolia* is

also much larger than 0.007–0.042 cm s$^{-1}$ range found for Norway spruce (*Picea abies)* by Breuninger et al. (2012), but surprisingly comparable, given the differences in plant species, to the 0.12 cm s$^{-1}$ deposition velocity found for maize (*Zea mays*) by Hereid and Monson (2001). We also find here a NO$_2$ flux at 5 ppb of 0.2 nmol m$^{-1}$ s$^{-1}$, similar in magnitude to the 0.1 nmol m$^{-1}$ s$^{-1}$, 0.15–1.5 nmol m$^{-1}$ s$^{-1}$, and 0.18 nmol m$^{-1}$ s$^{-1}$ fluxes measured for *Fagus sylvatica* (Gessler et al., 2000)*,* tropical Panamanian native trees (Sparks et al., 2001), and periwinkle (*Catharanthus roseus)* (Teklemariam and Sparks,

2006), respectively.

Resistance parameters reported above for NO deposition to *Quercus agrifolia* represent a stomatal deposition velocity of $0.007 \pm 0.002$ cm s$^{-1}$ and cuticular deposition velocity of $0.005 \pm 0.001$ cm s$^{-1}$. This observation of very minor NO uptake—at least an order of magnitude less than that of NO$_2$ uptake—is also consistent with previous observations (Hanson and Lindberg, 1991; Hereid and Monson, 2001; Teklemariam and Sparks, 2006). We also detected a statistically

significant NO compensation point, with low emissions up to 8 pmol m$^{-2}$ s$^{-1}$ observed below 1 ppb. These observations are similar to the 8–14 pmol m$^{-2}$ s$^{-1}$ emission fluxes of NO reported by Hereid and Monson (2001) and Teklemariam and Sparks (2006) at low NO$_x$ concentrations.

No significant NO$_2$ compensation point was found for our measurements of *Quercus agrifolia* NO$_x$ uptake. Many previous studies have reported NO$_2$ compensation points, ranging from 0.1–3.0 ppb, implicating trees as a constant source of

NO$_x$ in forest ecosystems (Gessler et al., 2000; Hereid and Monson, 2001; Sparks et al., 2001; Teklemariam and Sparks, 2006). Our findings of a lack of NO$_2$ compensation point support field observations and modeling studies that have recognized NO$_2$ dry deposition to vegetation as an important NO$_x$ loss process in forests (Jacob and Wofsy, 1990; Ganzeveld et al., 2002b; Geddes and Murphy, 2014). Our results also support the works of Chaparro-Suarez et al. (2011) and Breuninger et al. (2013), who did not find evidence of an NO$_2$ compensation point.

The primary difference in our experimental setup, compared to previous dynamic chamber studies that have found a NO$_2$ compensation point, is the use of a direct NO$_2$ measurement technique.   Measurements of a significant NO$_2$ compensation point have mostly been obtained using techniques requiring conversion of NO$_2$, followed by chemiluminescence detection of NO (Gessler et al., 2000; Hereid and Monson, 2001; Sparks et al., 2001; Teklemariam and





Sparks, 2006). Such methods have utilized either non-specific photolytic (Gessler et al., 2000; Hereid and Monson, 2001), luminol (Sparks et al., 2001), or catalytic conversion (Teklemariam and Sparks, 2006) techniques, which may have also resulted in the conversion of PAN, HONO, HNO$_3$, and other organic nitrates, as well as interferences from alkene + ozone reactions. If any of these interfering compounds are not excluded from the chamber system, or form from reactions of

biogenic emissions, this would cause an enhancement in observed NO$_2$ compensation point, and a suppression of observed deposition velocity. Additionally, previous chamber measurements have sometimes employed chamber setups that would let in a substantial amount of UV light, yet did not exclude photochemical reactions between NO$_2$, NO, and O$_3$. Such corrections are excluded here because of our use of chamber lights with only wavelengths above 420 nm. To avoid this issue, other experiments have instead involved a setup including a simultaneously measured blank chamber, which would

theoretically allow for correction for any reactions resulting from photolysis of NO$_2$, O$_2$, or O$_3$ (Gessler et al., 2000; Hereid and Monson, 2001). Such corrections might be complicated by secondary chemistry not present in our experiments.

### 4.2 Implication for canopy NO$_x$ loss

Resistance parameters reported above (Table 3) were used in a 1-D seven-layer multibox model representing chemical reactions, vertical transport, and leaf-level processes scaled to the canopy level to assess the impacts of NO$_x$ deposition

velocities on the NO$_x$ lifetime and fluxes. The model is constructed in a manner similar to Wolfe and Thornton (2011). Details will be presented elsewhere. The 1-D model was run for meteorological conditions representing the native habitat of *Quercus agrifolia* and two different leaf area indices (LAI) approximately representing the lower and upper limits of LAI found in California oak woodlands. As shown in Fig. 5a and 5b, the model predicts NO$_x$ deposition to *Quercus agrifolia* accounts for 3%–7% of the total NO$_x$ loss within the boundary layer if the only source of NO$_x$ is emissions from the soil.

This represents a total NO$_x$ lifetime of 7–7.5 hours in the boundary layer, and a lifetime to deposition of 4–11 days in the boundary layer and 0.5–1.2 hours below the canopy. Under these scenarios approximately 15–30% of soil-emitted NO$_x$ is removed in the canopy (Fig. 6)—on the lower end of the range of 25–80% reduction observed in field studies (Jacob and Wofsy, 1990; Lerdau et al., 2000; Ganzeveld et al., 2002a; Min et al., 2014).

     The coastal regions of California where *Q. agrifolia* is found frequently experience much higher NO$_x$ mixing ratios

of 10–50 ppb. This is particularly important for oak woodlands of the San Francisco Bay and near Los Angeles areas, where anthropogenic emissions from nearby urban centers are the majority of the NO$_x$ source. To account for this extra NO$_x$ source, additional model runs were done with an added term accounting for NO$_x$ advection from a more concentrated upwind source ($C_{NO_x(adv)}$), with advection treated as a simple mixing process:

$$\left(\frac{dC_{NO_x}}{dt}\right) = -k_{mix}\left(C_{NO_x} - C_{NO_x(adv)}\right) \tag{11}$$

where $k_{mix} = 0.3$ h$^{-1}$ and $C_{NO_x(adv)}$ is 10 ppb.



In this case, deposition to *Q. agrifolia* could account for 10–22% of the total $NO_x$ loss (Fig. 5c,d), representing a lifetime to deposition of 5–14 days in the boundary layer and a total $NO_x$ lifetime of 28–33 hours. Deposition in this higher $NO_x$ scenario decreased the total $NO_x$ lifetime by 3–8 hours, compared with a no-deposition case.

## 5 Conclusions

This work constitutes the first measurements of $NO_2$ and NO foliar deposition resistance parameters for a North American tree species. We report observations of leaf-level resistances to $NO_2$ and NO deposition, corresponding to total deposition velocities of $NO_2$ and NO of $0.123 \pm 0.007$ cm s$^{-1}$ and $0.012 \pm 0.002$ cm s$^{-1}$ in the light and $0.015 \pm 0.001$ cm s$^{-1}$ and $0.005 \pm 0.002$ cm s$^{-1}$ in the dark, respectively. No compensation point was observed for $NO_2$, but compensation points of 0.74–3.8 ppb were recorded for NO. The magnitude of NO emission below the compensation point was significantly less

than the magnitude of $NO_2$ uptake in the same concentration range, making *Q. agrifolia* an overall large net sink of $NO_x$. The observed deposition is large enough to explain canopy reduction factors observed in canopy-level studies, but is at the lower end of estimated global CRFs. The results of the 1-D multibox model demonstrate that the deposition observed accounts for 5–20% of $NO_x$ removal with a $NO_x$ lifetime to deposition of 0.5–1.2 hours beneath the canopy of a California oak woodland. We show that foliar deposition of $NO_x$ represents a significant removable mechanism of $NO_x$ and can have a large impact on

$NO_x$ mixing ratios and fluxes in such ecosystems. Further investigations of $NO_2$ deposition to a larger variety of plant species under a range of environmental conditions are needed to accurately understand the global impacts of $NO_2$ deposition across diverse ecosystems.

*Acknowledgements.* The authors wish to gratefully acknowledge financial support from the National Science Foundation (NSF, AGS-1352972). Additional support was provided by a NSF Graduate Research Fellowship to Erin R. Delaria.



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



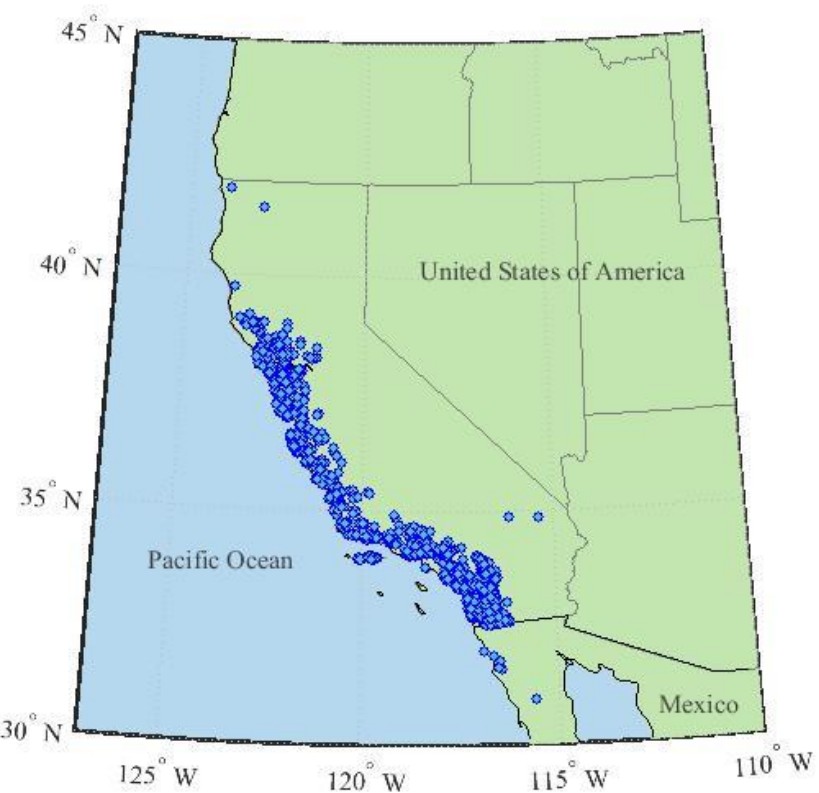

**Figure 1: Species distribution map of *Quercus agrifolia*. Each dot represents an observation of *Q. agrifolia* occurrence. Data provided by the participants of the Consortium of California Herbaria.**


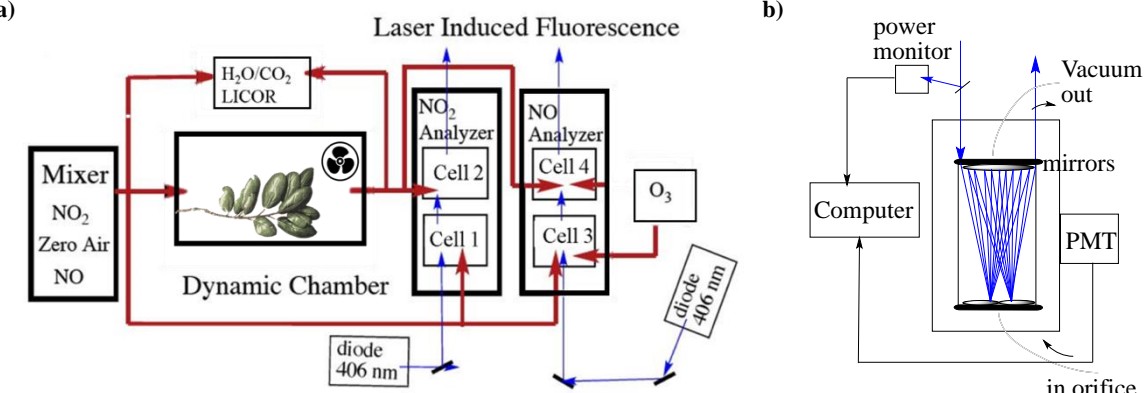

**Figure 2: Schematic of the experimental dynamic chamber (a) and laser-induced fluorescence detection (b) setups.**




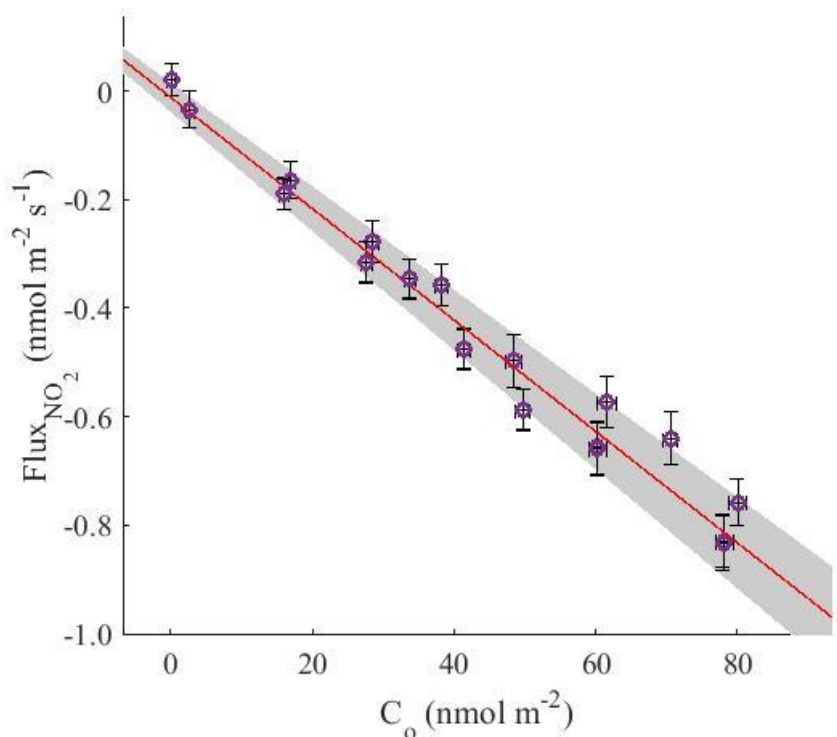

**Figure 3: Flux to a 5.1 cm diameter dish filled with activated charcoal. The chemical surface resistance to deposition is approximately zero, so the deposition velocity for deposition of NO₂ to the surface of the charcoal dish is the reciprocal of the boundary layer resistance.The line of best fit is $(0.51 \pm 0.032)C_o$, where $C_o$ is the concentration of NO₂ in the outgoing airstream.**



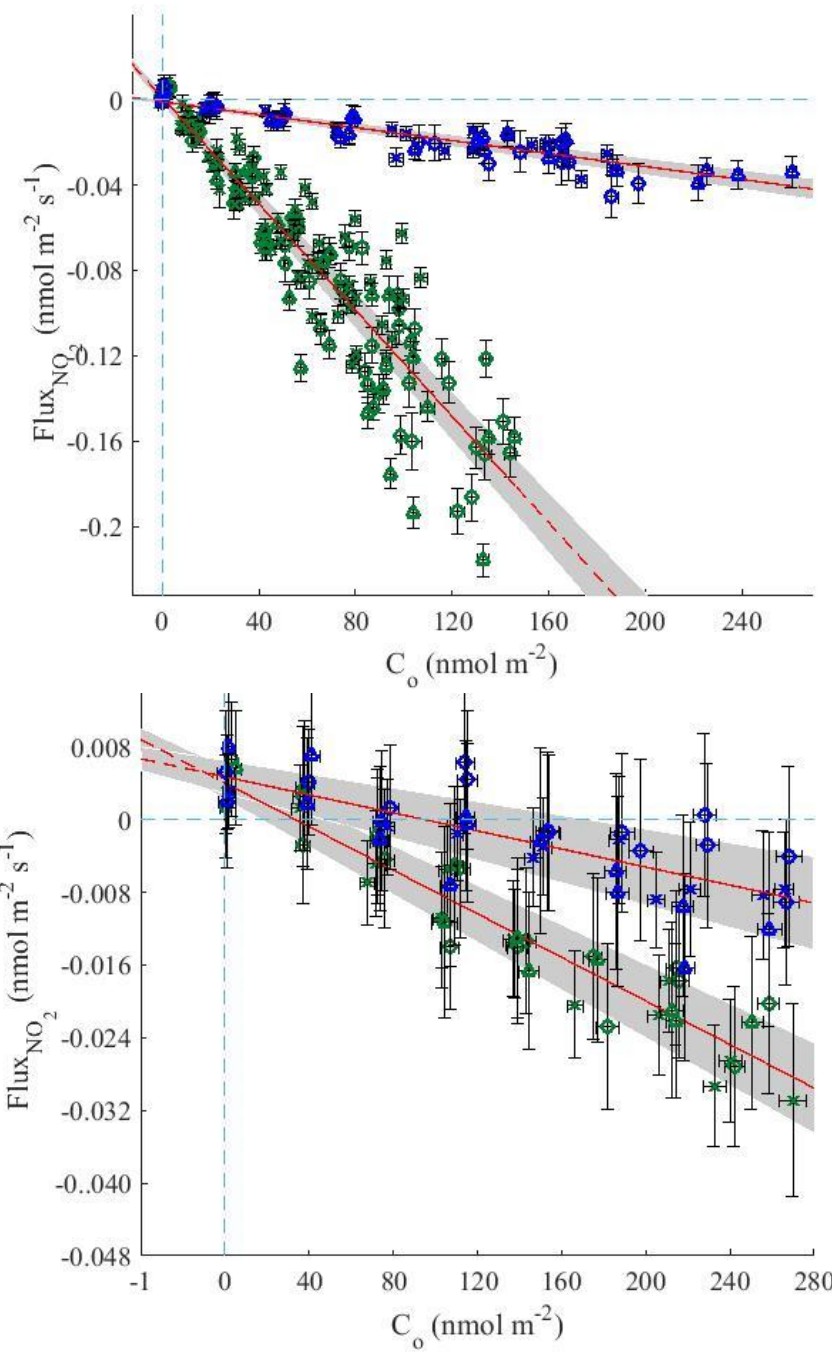

**Figure 4: NO₂ (top) and NO (bottom) fluxes versus the outlet concentrations for all *Quercus agrifolia* individuals with the chamber lights on (green) and off (blue). The line of best fit is shown in red and was calculated to minimize the weighted residuals in both the x- and y- axis. The blue dotted line shows where flux is zero. A significantly positive ($\alpha = 0.5$) x-intercept occurs for NO, but not NO₂ experiments.**





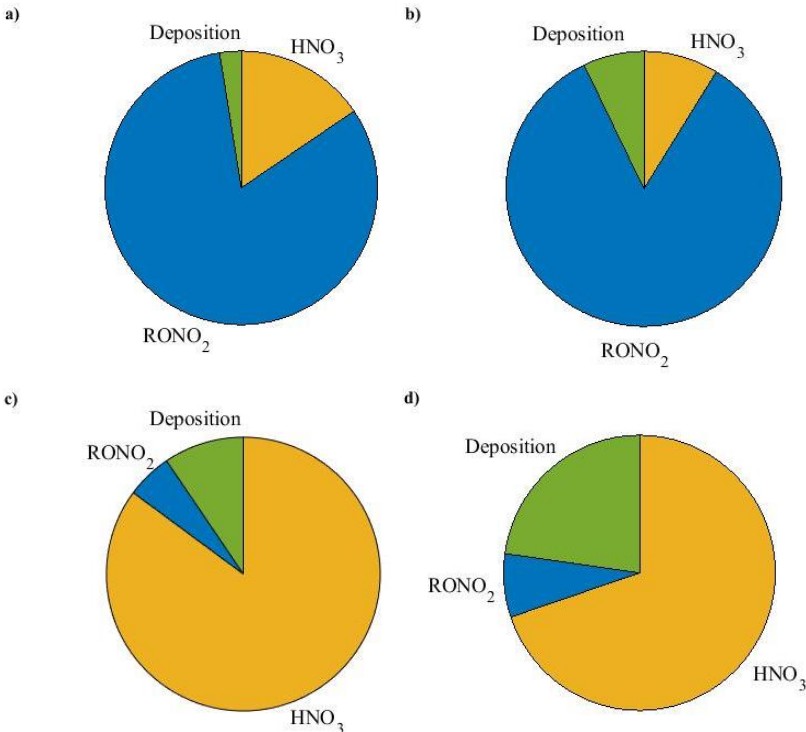

Figure 5: Model predictions of the fraction of NO$_x$ loss to alkyl nitrate formation, nitric acid formation, and deposition in a *Q. agrifolia* woodland. The model was run using scenarios with only soil emissions and LAI of 1 m²/m² (a), only soil emissions and LAI of 3 m²/m² (b), $C_{NO_x(adv)} = 10$ ppb and LAI of 1 m²/m² (c), and $C_{NO_x(adv)} = 10$ ppb and LAI of 3 m²/m² (d).





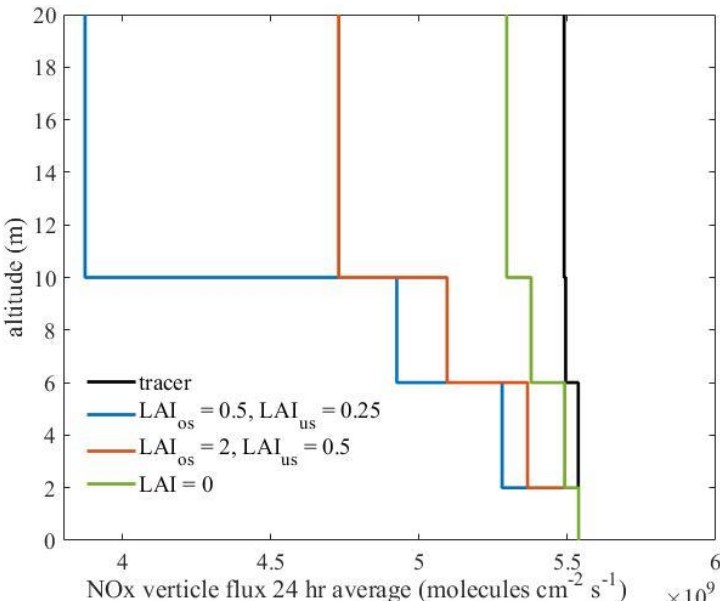

5    **Figure 6: 24 hr average vertical fluxes of NOₓ predicted by the 1-D multibox model for a California oak woodland using the leaf resistances measured in this study. Model runs were conducted for a low (red) and high (blue) LAI cases and for a no deposition scenario (green).**



**Table 1: Parameters of NO₂ bi-variate linear least-square fitting regression analysis**

| run | N | $R^2$ | $[NO_2]_{comp}$ (ppb) | $P([NO_2]_{comp}=0)$ % | $V_{dep}$ cm s⁻¹ |
|---|---|---|---|---|---|
| *Q.agrifolia* 1, light | | | | | |
| 1 | 13 | 0.979 | $0.056 \pm 0.013$ | 42.7 | $0.10 \pm 0.013$ |
| 2 | 13 | 0.950 | $0.046 \pm 0.19$ | 63.7 | $0.12 \pm 0.023$ |
| 3 | 16 | 0.978 | $0.099 \pm 0.086$ | 3.87 | $0.15 \pm 0.016$ |
| 4 | 16 | 0.958 | $0.077 \pm 0.14$ | 28.7 | $0.12 \pm 0.021$ |
| All | 58 | 0.927 | $0.080 \pm 0.10$ | 11.6 | $0.12 \pm 0.012$ |
| *Q.agrifolia* 2, light | | | | | |
| 1 | 16 | 0.963 | $0.10 \pm 0.12$ | 10.3 | $0.08 \pm 0.011$ |
| 2 | 5 | 0.969 | $-0.01 \pm 0.96$ | 83.8 | $0.12 \pm 0.014$ |
| 3 | 9 | 0.997 | $0.023 \pm 0.032$ | 20.3 | $0.16 \pm 0.011$ |
| 4 | 16 | 0.9736 | $-0.019 \pm 0.074$ | 61.9 | $0.14 \pm 0.017$ |
| 5 | 15 | 0.979 | $0.015 \pm 0.082$ | 72.7 | $0.12 \pm 0.014$ |
| All | 71 | 0.845 | $-0.0077 \pm 0.091$ | 91.6 | $0.11 \pm 0.014$ |
| *Q.agrifolia* 3, light | | | | | |
| 1 | 11 | 0.969 | $0.016 \pm 0.18$ | 87.4 | $0.12 \pm 0.024$ |
| 2 | 15 | 0.961 | $0.074 \pm 0.16$ | 39.1 | $0.18 \pm 0.029$ |
| 3 | 5 | 0.990 | $0.30 \pm 0.20$ | 5.9 | $0.12 \pm 0.038$ |
| All | 31 | 0.830 | $0.019 \pm 0.064$ | 77.6 | $0.14 \pm 0.029$ |
| All *Q.agrifolia*, light | 160 | 0.885 | $0.030 \pm 0.072$ | 41.3 | $0.123 \pm 0.0092$ |
| *Q.agrifolia* 1, dark | | | | | |
| 1 | 16 | 0.964 | $0.056 \pm 0.14$ | 0.9* | $0.022 \pm 0.0034$ |
| *Q.agrifolia* 2, dark | | | | | |
| 1 | 16 | 0.858 | $-0.16 \pm 0.47$ | 50.8 | $0.016 \pm 0.0050$ |
| 2 | 12 | 0.932 | $-0.34 \pm 0.40$ | 11.8 | $0.013 \pm 0.0038$ |
| All | 28 | 0.853 | $-0.24 \pm 0.32$ | 15.6 | $0.015 \pm 0.0030$ |
| *Q.agrifolia* 3, dark | | | | | |
| 1 | 14 | 0.900 | $-0.30 \pm 0.48$ | 24.1 | $0.015 \pm 0.0042$ |
| 2 | 11 | 0.909 | $-0.001 \pm 0.69$ | 36.7 | $0.015 \pm 0.0057$ |
| All | 25 | 0.898 | $-0.22 \pm 0.38$ | 25.3 | $0.014 \pm 0.0029$ |
| All *Q.agrifolia*, dark | 69 | 0.881 | $-0.16 \pm 0.24$ | 12.2 | $0.015 \pm 0.0018$ |

**\* Significant non-zero compensation point**



**Table 2: Parameters of NO bi-variate linear least-square fitting regression analysis**

| run | N | $R^2$ | $[NO_2]_{comp}$ (ppb) | $P([NO_2]_{comp}=0)$ | $V_{dep}$ |
|---|---|---|---|---|---|
| *Q. agrifolia* 1 | | | | | |
| light | 17 | 0.874 | $0.74 \pm 0.65$ | 3.5* | $0.011 \pm 0.0032$ |
| dark | 13 | 0.699 | $3.8 \pm 2.2$ | 0.52* | $0.0040 \pm 0.0025$ |
| *Q. agrifolia* 1 | | | | | |
| light | 14 | 0.954 | $0.76 \pm 0.49$ | 0.92* | $0.013 \pm 0.0027$ |
| dark | 10 | 0.866 | $1.7 \pm 1.0$ | 1.1* | $0.0046 \pm 0.0018$ |
| *Q. agrifolia* 1 | | | | | |
| light | 12 | 0.936 | $1.3 \pm 0.60$ | 0.17* | $0.0123 \pm 0.0029$ |
| dark | 15 | 0.803 | $2.0 \pm 1.0$ | 2.5* | $0.0074 \pm 0.0033$ |
| All *Q. agrifolia* | | | | | |
| light | 13 | 0.908 | $0.84 \pm 0.32$ | <0.01* | $0.012 \pm 0.0015$ |
| dark | 13 | 0.602 | $2.4 \pm 1.1$ | <0.01* | $0.0050 \pm 0.0016$ |

**\*Significant non-zero compensation point**



**Table 3: Summary of deposition resistance parameters of *Quercus agrifolia***

| gas | $R_b$ | $R_{cut}$ | $R_{st} + R_m$ |
| --- | --- | --- | --- |
| | s cm$^{-1}$ | s cm$^{-1}$ | s cm$^{-1}$ |
| NO$_2$ | $1.94 \pm 0.02$ | $63 \pm 8$ | $6.9 \pm 0.9$ |
| NO | $2.59 \pm 0.03$ | $200 \pm 60$ | $140 \pm 40$ |