# Peer review of "Measurements of NO and NO2 exchange between the atmosphere and *Quercus agrifolia"

_Atmospheric Chemistry and Physics, 2018_

## Referee Comment (RC1) · Anonymous Referee #1 · 29 Jun 2018

Review of manuscript acp-2018-433, entitled "Measurements of NO and NO2 exchange between the atmosphere and Quercus agrifolia," by Delaria et al.

This paper presents results from laboratory studies of the fluxes of nitric oxide (NO) and nitrogen dioxide (NO2) over California oak using a branch enclosure technique. The paper is generally well written and cites relevant previous work in this field. The experimental method is sound and the results are in general accord with previous studies.

This paper is a sound contribution to this field of research and should be published, but only after addressing the comments I present below.

1. The main problem I have with this paper is the focus by the authors on making com-

[Figure]

parisons of the LIF instrument that they use to measure NO2 to other techniques. This paper is not about comparison of measurement techniques, nor do they present anything new in that regard. A fair amount of text is devoted to pointing out interferences, especially with the photolytic/chemiluminescence (CL) NO2 technique. Not only are these comparisons unnecessary and distracting, but these points already have been made in prior literature.

Two examples will suffice here. The first is the authors' argument about detection limits in techniques other than LIF being insufficient to measure the low mixing ratios observed in laboratory and field measurements of fluxes. This is a specious argument since the mixing ratios used in this study (0.5 - 10 ppb of NO and NO2 - see abstract) are well above detection limits of the other techniques the authors question. And, again, those other techniques were not tested in the present study. The second point is the argument about ozone-alkene reactions causing interference in the CL method due to possible high levels of biogenic alkenes being emitted by vegetation and causing measurement interference when reacted with ozone reagent gas in the CL technique. This is well-documented in the literature and has been shown to be negligible with modern CL instruments. I would point out that if this is a significant effect, it may be an interference in this work for NO since excess ozone was added to the flux chamber to convert NO to NO2 for measurement by LIF, which uses similar red-sensitive photomultiplier tubes to the CL method.

The point here is not to place doubt on the LIF method, but to remove unnecessary and distracting text from the paper. Plus, shorter is better for most papers.

2. I have one question on the presentation of the results. It seems to me that when mean deposition velocities or resistances are shown, the uncertainty is understated. For example, in Table 1 for all NO2 deposition velocities under lighted conditions, the mean is 0.12 +/- 0.012. I can see how this was calculated, but I wonder if the listed uncertainty is the most appropriate value or the one most useful to the community. It seems to me that each Vdep should be calculated as an independent value and those

averaged together to give a more meaningful estimate of actual variability. Can the authors comment on this?

3. I found some typographical errors that should be corrected. Fig. 3: C0 should be nmol m-3 Fig. 4: C0 in both plots should be nmol m-3 and bottom plot y-axis should be NO Fig. 5: verticle in x-axis label

---

## Referee Comment (RC2) · L. Ganzeveld (Referee) · 7 Aug 2018

Review of paper acp-2018-433: Measurements of NO and NO2 exchange between the atmosphere and Quercus agrifolia by Delaria et al.

The paper describes analysis of a measurement dataset on NO2 and NO exchange flux measurements using enclosure experiments on Quercus agrifolia. This is followed by interpretation of the established dry deposition velocities and NO compensation point for the overall contribution by deposition to these tree species for boundary layer NOx using a simple multi-box modelling system. Overall, this reads as a nice comprehensive study clearly indicating the implications of the results found in these experiments. Consequently, I do recommend publication of this study that does not only report on the

exchange characteristics of NOx for this specific tree species but also addresses potential issues involved in previous studies on NOx compensation points. I have mainly some minor comments mostly focussing on some of the modelling features and which hopefully nicely complements the comments by the other reviewer who had more specific comments regarding the experimental component of the study.

Below you can find my more specific comments.

Pp2, line 16: in this statement about the use the CRF referring to Ganzeveld et al. it is suggested that in this study the CRF was applied to correct the soil NO emissions. This is actually not the case; that study used a multi-layer exchange model to explicitly calculate the effective exchange between the canopy and the atmosphere and which yielded a canopy-top to soil NO emission flux quite comparable to the CRF proposed by Yienger and Levy of 50% for tropical forests. By the way, the study by Ganzeveld et al. (2002a) also presented a sensitivity analysis regarding the significance of this NO2 compensation point for global scale atmosphere-biosphere NOx exchange.

Pp 3, line 7: "...uptake rates necessary to describe the observed 20–50% reduction of soil-emitted NOx...". This statement suggests that the 20-50% of reduction of soil NO emissions can be completely explained by the NO2 removal rate. It is indeed true that existing models of in-canopy NOx cycling suggest that these canopy reduction factors are dominated by VdNO2 but we can also not rule out the important role of gradients in photolysis effecting the gradients and, consequently, atmosphere-biosphere fluxes and other in-canopy chemical transformations/interactions.

Pp3, line 17: "Observations of NOx canopy fluxes and atmospheric models..."; here you suggest that model studies show that trees take up NOx mixing ratios over 0.1 ppbv. What atmospheric models are those?? I think that models generally produce a whole range of results on NOx fluxes dependent on how the biogenic emissions, dry deposition (and canopy interactions) have been implemented and on the assumptions being made but which up to now lack actually lots of experimental information on issues

such as the existence of the compensation point. Here we really need to connect leaf-to canopy-scale and in-canopy NOx gradient and flux measurements together with multi-layer exchange models to further demonstrate the potential existence and relevance of leaf- to canopy-scale NOx compensation points for difference ecosystems.

Page 5, line 19: "This corresponded to a maximum loss of 0.4 ppb at 8 ppb NO2". Can you assume that the wall loss scales linearly with the concentration? What are the wall losses for the minimum concentrations you used for the experiments?

Page 8-9: "For all light and dark experiments the average compensation point for NO was calculated as $0.84 \pm 0.32$ ppb NO and $2.4 \pm 1.1$ ppb NO, respectively (Table 2).

Page 9: "making Quercus agrifolia a large net sink of NOx"; I see here your point that this tree species seems to be a sink of NOx given that the NO emission flux is only half the NO2 deposition flux but this doesn't confirm so much that this tree species is overall providing a large sink of NOx (which would depend on the overall functioning of the canopy–soil system). Reading over then also later on Section 4.2, this is indeed confirmed having an overall loss by deposition to these trees on the order of 3-7% of total NOx loss in the boundary layer.

Regarding the presented study on the implications of the leaf-level measurements of NO2/NO compensation points for canopy-scale NOx exchange, there is a study by Seok et al. (Dynamics of nitrogen oxides and ozone above and within a mixed hard-wood forest in northern Michigan ACP, 2013) that addressed the potentially important role of the compensation point based on analysis of in and above-canopy NOx concentration dynamics also using a multi-layer exchange model on this dataset. The observed early morning peak of NOx was best explained actually considering the role of a NOx compensation point in the exchange simulations.

---

## Author Comment (AC1) · 14 Sep 2018

The following addresses comments of both reviewers and outlines revisions made to the manuscript.

We are very grateful for the constructive comments and valuable suggestions offered by the two reviewers. The reviewers' comments appear in *blue italics*, followed by our responses to each comment in plain black text. Line numbers refer to the original, unedited manuscript document version.

**Reviewer 1**

*C1) The main problem I have with this paper is the focus by the authors on making com-*

*parisons of the LIF instrument that they use to measure NO2 to other techniques. This paper is not about comparison of measurement techniques, nor do they present anything new in that regard. A fair amount of text is devoted to pointing out interferences, especially with the photolytic/chemiluminescence (CL) NO2 technique. Not only are these comparisons unnecessary and distracting, but these points already have been made in prior literature.Two examples will suffice here. The first is the authors' argument about detection limits in techniques other than LIF being insufficient to measure the low mixing ratios observed in laboratory and field measurements of fluxes. This is a specious argument since the mixing ratios used in this study (0.5 - 10 ppb of NO and NO2 - see abstract) are well above detection limits of the other techniques the authors question. And, again, those other techniques were not tested in the present study. The second point is the argument about ozone-alkene reactions causing interference in the CL method due to possible high levels of biogenic alkenes being emitted by vegetation and causing measurement interference when reacted with ozone reagent gas in the CL technique. This is well-documented in the literature and has been shown to be negligible with modern CL instruments. I would point out that if this is a significant effect, it may be an interference in this work for NO since excess ozone was added to the flux chamber to convert NO to NO2 for measurement by LIF, which uses similar redsensitive photomultiplier tubes to the CL method. The point here is not to place doubt on the LIF method, but to remove unnecessary and distracting text from the paper. Plus, shorter is better for most papers.*

We removed lines 17-30 on page 3, and moved lines 5-7 on page 5, beginning: "Our measurements..." to the discussion section on line 6 of pg. 11. We also moved the phrase at the end of the same paragraph on page 5 :"...at low $NO_x$ mixing ratios relevant to forested environments" to the end of the sentence, which begins line 34 pg. 3, so it reads: "With the LIF technique we are able measure $NO_x$ exchange fluxes with high specificity and sensitivity at low $NO_x$ mixing ratios relevant to forested environments." Discussion of chemiluminescence measurements is limited to section 4.1 on pg 11, where differences between $NO_2$ deposition results of our study and previous

measurements are discussed.

*C2) I have one question on the presentation of the results. It seems to me that when mean deposition velocities or resistances are shown, the uncertainty is understated. For example, in Table 1 for all NO2 deposition velocities under lighted conditions, the mean is 0.12 +/- 0.012. I can see how this was calculated, but I wonder if the listed uncertainty is the most appropriate value or the one most useful to the community. It seems to me that each Vdep should be calculated as an independent value and those averaged together to give a more meaningful estimate of actual variability. Can the authors comment on this?*

The method suggested by the reviewer results in a similar value (mean of $0.128 \pm 0.015$), when the Vdep from each light $NO_2$ experiment is averaged and the error calculated. However, the value of $0.123 \pm 0.0099$ is obtained if a weighted mean (weighted by error in the individual Vdeps) and standard deviation is calculated from the individual Vdeps (which is quite similar to the $0.123 \pm 0.0092$ cm/s reported in Table 1). We argue that the weighted mean is a more accurate representation of the Vdep mean and error. The mean and confidence interval reported was determined from a regression of all data from all light or dark experiments, which essentially calculates a weighted mean of all Vdeps of each individual data point. The range of all measured Vdep is included to the text to reflect the range of Vdep measured, and may be used as a more conservative estimate of the uncertainty.

We also corrected line 6 of page 9 to read "...deposition of $0.123 \pm 0.009$ cm/s...", in agreement with the value reported in Table 1.

*C3) I found some typographical errors that should be corrected. Fig. 3: C0 should be nmol m-3 Fig. 4: C0 in both plots should be nmol m-3 and bottom plot y-axis should be NO Fig. 5: verticle in x-axis label.*

The errors pointed out in Fig. 3, Fig. 4, And Fig. 6 were corrected. I believe the reviewer comment referencing Fig. 5 was discussing Fig. 6.

**Reviewer 2**

*C1) Pp2, line 16: in this statement about the use the CRF referring to Ganzeveld et al. It is suggested that in this study the CRF was applied to correct the soil NO emissions. This is actually not the case; that study used a multi-layer exchange model to explicitly calculate the effective exchange between the canopy and the atmosphere and which yielded a canopy-top to soil NO emission flux quite comparable to the CRF proposed by Yienger and Levy of 50% for tropical forests. By the way, the study by Ganzeveld et al. (2002a) also presented a sensitivity analysis regarding the significance of this NO2 compensation point for global scale atmosphere-biosphere NOx exchange.*

We revise the manuscript introduction so that citation of Ganzeveld et al. (2002a) is removed as a citation for the statement ending on line 16 of pg 2.

*C2) Pp 3, line 7: . . .uptake rates necessary to describe the observed 20–50% reduction of soil-emitted NOx. . .". This statement suggests that the 20-50% of reduction of soil NO emissions can be completely explained by the NO2 removal rate. It is indeed true that existing models of in-canopy NOx cycling suggest that these canopy reduction factors are dominated by VdNO2 but we can also not rule out the important role of gradients in photolysis effecting the gradients and, consequently, atmosphere-biosphere fluxes and other in-canopy chemical transformations/interactions.*

We revise the manuscript introduction (page 3, line 6 ) to better reflect the complexity of canopy reduction and the state of the current uncertainties: "Many laboratory experiments have failed to measure uptake rates necessary to describe the observed 20–50% reduction of soil-emitted $NO_x$ (Hanson and Lindberg, 1991; Breuninger et al., 2013), while many modeling studies have suggested dry deposition makes up most of this reduction (Jacob and Wofsy, 1990; Yienger and Levy, 1995; Ganzeveld et al., 2002a; Geddes and Murphy, 2014). Photolysis gradients and reaction of $NO_x$ to form higher nitrogen oxides may also account for a large fraction of this reduction in soil

NO$_x$, as has been suggested by Min et al. (2014, 2012), but the relative importance of dry deposition processes versus in-canopy chemical transformations is still a matter of considerable uncertainty (Lerdau et al., 2000; Ganzeveld et al., 2002a)."

*C3) Pp3, line 17: "Observations of NOx canopy fluxes and atmospheric models...";*
*here you suggest that model studies show that trees take up NOx mixing ratios over*
*0.1 ppbv. What atmospheric models are those?? I think that models generally produce*
*a whole range of results on NOx fluxes dependent on how the biogenic emissions, dry*
*deposition (and canopy interactions) have been implemented and on the assumptions*
*being made but which up to now lack actually lots of experimental information on issues*
*such as the existence of the compensation point. Here we really need to connect leaf-to*
*canopy-scale and in-canopy NOx gradient and flux measurements together with multi-*
*layer exchange models to further demonstrate the potential existence and relevance of*
*leaf- to canopy-scale NOx compensation points for difference ecosystems.*

What was meant in stating, "...trees are substantial sinks..." is that there is still dry deposition of NO$_2$ at even low NO$_x$ mixing ratios, not that the overall canopy flux is negative. The biogenic emissions, dry deposition, and canopy interactions could still, of course, make the forest system a net source of NO$_x$. The line was edited to reflect this. A few sentences were also added to address modelling studies (eg Seok et al., 2013) that have suggested the existence of a compensation point: "Emission of NO at these low NO$_x$ mixing ratios has also been detected in laboratory chamber studies (Wildt et al., 1997; Hereid and Monson, 2001). More recent laboratory studies of leaf level deposition have, however, questioned the existence of a compensation point (Chaparro-Suarez et al., 2011; Breuninger et al., 2013). Most observations of NO$_x$ canopy fluxes and atmospheric models predict or assume substantial NO$_x$ deposition at concentrations as low as 0.1 ppb, typical of NO$_x$ mixing ratios in remote areas (Jacob and Wofsy, 1990; Wang and Leuning, 1998; Lerdau et al., 2000; Sparks et al., 2001; Wolfe and Thornton, 2011; Min et al., 2012; Geddes and Murphy, 2014). However, some modelling studies have suggested that a NO$_2$ compensation point is necessary

to describe (Seok et al., 2013), or has only a small effect on canopy fluxes in most regions (Ganzeveld et al., 2002a). More research is thus needed on leaf and canopy-level processes to understand the full complexity of the soil-canopy-atmosphere system."

*C4) Page 5, line 19: "This corresponded to a maximum loss of 0.4 ppb at 8 ppb NO2". Can you assume that the wall loss scales linearly with the concentration? What are the wall losses for the minimum concentrations you used for the experiments?*

The wall loss had been directly measured and has been found to scale linearly and has already been subtracted from reported fluxes. The minimum concentration used was 1 ppb, corresponding to a wall loss of 0.05 ppb. We revised the manuscript to also state this minimum loss.

*C5) Page 8-9: "For all light and dark experiments the average compensation point for NO was calculated as 0.84 ± 0.32 ppb NO and 2.4 ± 1.1 ppb NO, respectively (Table 2). Page 9: "making Quercus agrifolia a large net sink of NOx"; I see here your point that this tree species seems to be a sink of NOx given that the NO emission flux is only half the NO2 deposition flux but this doesn't confirm so much that this tree species is overall providing a large sink of NOx (which would depend on the overall functioning of the canopy–soil system). Reading over then also later on Section 4.2, this is indeed confirmed having an overall loss by deposition to these trees on the order of 3-7% of total NOx loss in the boundary layer.*

We revised the manuscript page 9 so that the sentence the reviewer referred to specifies that the deposition process, specifically, acts as a sink, not necessarily the canopy-soil system as a whole: "At typical $NO_2/NO$ ratios and gradients measured in forest canopies, the leaf-level $NO_2$ and NO exchange fluxes measured make dry stomatal deposition to Quercus agrifolia a net sink of $NO_x$ within the canopy".

*C6) Regarding the presented study on the implications of the leaf-level measurements of NO2/NO compensation points for canopy-scale NOx exchange, there is a study by Seok et al. (Dynamics of nitrogen oxides and ozone above and within a mixed hard-*

*wood forest in norther Michigan ACP, 2013) that addressed the potentially important role of the compensation point based on analysis of in and above-canopy NOx concentration dynamics also using a multi-layer model on this dataset. The observed early morning peak of NOx was best explained actually considering the role of a NOx compensation point in the exchange simulations.*

We agree with the reviewer that the study by Seok et al. is quite relevant to the implications for the compensation point discussed in this paper. Seok et al. 2013 was added as a reference in edits made while addressing C3.
* * *
We also made the following additional edits to the manuscript:

1) An error was found in the calculation of the $NO_2$ cuticular resistance, which was corrected in the text and Table 3. The cuticular resistance is corrected to $65 \pm 8$ s cm$^{-1}$.

2)Statements were added to address possible errors in the determination of the boundary resistance.

The statement added to line 29 on page 7 (section 3.1) reads: "However, with a branch enclosed inside the chamber, the effective boundary resistance to deposition will likely be reduced, as the surface roughness and surface area for deposition is increased (Galbally and Roy, 1980; Pape et al., 2009). The boundary resistances presented above thus serve as an upper limit for Rb with vegetation inside the chamber."

The statement added to line 22 on page 9 (section 3.3) reads: " It should be noted that since the reported $R_b$ is the maximum possible boundary resistance, the reported $R_{cut}$ and $R_s^*$ are lower limits. If we were to assume the chamber boundary resistance with the branch enclosed is insignificant ( 0 s cm$^{-1}$), this would introduce maximum systematic 30% and 3% errors to the calculated $NO_2$ $R_s^*$ and $R_{cut}$, respectively (giving an $R_s^*$ of $9.2 \pm 0.9$ s cm$^{-1}$ and an $R_{cut}$ of $67 \pm 8$ s cm$^{-1}$). The errors in the calculated NO resistances would be negligible."

3) We corrected minor typographical and grammatical errors.

---

## Author Response (AR1)

**Response to Reviewers**

We are very grateful for the constructive comments and valuable suggestions offered by the two reviewers. The reviewers' comments appear in *blue italics,* followed by our responses to each comment in plain black text. Line numbers refer to the original, unedited manuscript document version.

**Reviewer #1:**

*C1) The main problem I have with this paper is the focus by the authors on making comparisons of the LIF instrument that they use to measure NO2 to other techniques. This paper is not about comparison of measurement techniques, nor do they present anything new in that regard. A fair amount of text is devoted to pointing out interferences, especially with the photolytic/chemiluminescence (CL) NO2 technique. Not only are these comparisons unnecessary and distracting, but these points already have been made in prior literature. Two examples will suffice here. The first is the authors' argument about detection limits in techniques other than LIF being insufficient to measure the low mixing ratios observed in laboratory and field measurements of fluxes. This is a specious argument since the mixing ratios used in this study (0.5 - 10 ppb of NO and NO2 - see abstract) are well above detection limits of the other techniques the authors question. And, again, those other techniques were not tested in the present study. The second point is the argument about ozone-alkene reactions causing interference in the CL method due to possible high levels of biogenic alkenes being emitted by vegetation and causing measurement interference when reacted with ozone reagent gas in the CL technique. This is well-documented in the literature and has been shown to be negligible with modern CL instruments. I would point out that if this is a significant effect, it may be an interference in this work for NO since excess ozone was added to the flux chamber to convert NO to NO2 for measurement by LIF, which uses similar red-sensitive photomultiplier tubes to the CL method. The point here is not to place doubt on the LIF method, but to remove unnecessary and distracting text from the paper. Plus, shorter is better for most papers.*

We removed lines 17-30 on page 3, and moved lines 5-7 on page 5, beginning: "Our measurements…" to the discussion section on line 6 of pg. 11. We also moved the phrase at the end of the same paragraph on page 5 :"…at low $NO_x$ mixing ratios relevant to forested environments" to the end of the sentence, which begins line 34 pg. 3, so it reads: "With the LIF technique we are able measure NOx exchange fluxes with high specificity and sensitivity  at low NOx mixing ratios relevant to forested environments." Discussion of chemiluminescence measurements is limited to section 4.1 on pg 11, where differences between $NO_2$ deposition results of our study and previous measurements are discussed.

*C2) I have one question on the presentation of the results. It seems to me that when mean deposition velocities or resistances are shown, the uncertainty is understated. For example, in Table 1 for all NO2 deposition velocities under lighted conditions, the mean is 0.12 +/- 0.012. I can see how this was calculated, but I wonder if the listed uncertainty is the most appropriate value or the one most useful to the community. It seems to me that each Vdep should be calculated as an independent value and those averaged together to give a more meaningful estimate of actual variability. Can the authors comment on this?*

The method suggested by the reviewer results in a similar value (mean of 0.128 +/- 0.015), when the Vdep from each light $NO_2$ experiment is averaged and the error calculated. However, the value of 0.123 +/- 0.0099 is obtained if a weighted mean (weighted by error in the individual Vdeps) and standard deviation is calculated from the individual Vdeps (which is quite similar to the $0.123 \pm 0.0092$ cm/s reported in Table 1). We argue that the weighted mean is a more accurate representation of the Vdep mean and error. The mean and confidence interval reported was determined from a regression of all data from all light or dark experiments, which essentially calculates a weighted mean of all Vdeps of each individual data point. The range of all measured Vdep is included to the text to reflect the range of Vdep measured, and may be used as a more conservative estimate of the uncertainty.

We also corrected line 6 of page 9 to read "…deposition of $0.123 \pm 0.009$ cm/s…", in agreement with the value reported in Table 1.

*C3) I found some typographical errors that should be corrected. Fig. 3: C0 should be nmol m-3 Fig. 4: C0 in both plots should be nmol m-3 and bottom plot y-axis should be NO Fig. 5: verticle in x-axis label.*

The errors pointed out in Fig. 3, Fig. 4, And Fig. 6 were corrected. I believe the reviewer comment referencing Fig. 5 was discussing Fig. 6.

**Reviewer #2**

*C1) Pp2, line 16: in this statement about the use the CRF referring to Ganzeveld et al. It is suggested that in this study the CRF was applied to correct the soil NO emissions. This is actually not the case; that study used a multi-layer exchange model to explicitly calculate the effective exchange between the canopy and the atmosphere and which yielded a canopy-top to soil NO emission flux quite comparable to the CRF proposed by Yienger and Levy of 50% for tropical forests. By the way, the study by Ganzeveld et al. (2002a) also presented a sensitivity analysis regarding the significance of this NO2 compensation point for global scale atmosphere-biosphere NOx exchange.*

We revise the manuscript introduction so that citation of Ganzeveld et al. (2002a) is removed as a citation for the statement ending on line 16 of pg 2.

*C2) Pp 3, line 7: ...uptake rates necessary to describe the observed 20–50% reduction of soil-emitted NOx…". This statement suggests that the 20-50% of reduction of soil NO emissions can be completely explained by the NO2 removal rate. It is indeed true that existing models of in-canopy NOx cycling suggest that these canopy reduction factors are dominated by VdNO2 but we can also not rule out the important role of gradients in photolysis effecting the gradients and, consequently, atmosphere-biosphere fluxes and other in-canopy chemical transformations/interactions.*

We revise the manuscript introduction (page 3, line 6 ) to better reflect the complexity of canopy reduction and the state of the current uncertainties: "Many laboratory experiments have failed to measure uptake rates necessary to describe the observed 20–50% reduction of soil-emitted NOx (Hanson and Lindberg, 1991; Breuninger et al., 2013), while many modeling studies have suggested dry deposition makes up most of this reduction (Jacob and Wofsy, 1990; Yienger and Levy, 1995; Ganzeveld et al., 2002a; Geddes and Murphy, 2014). Photolysis gradients and reaction of NOx to form higher nitrogen oxides may also account for a large fraction of this reduction in soil NOx, as has been suggested by Min et al. (2014, 2012), but the relative importance of dry deposition processes versus in-canopy chemical transformations is still a matter of considerable uncertainty (Lerdau et al., 2000; Ganzeveld et al., 2002a)."

*C3) Pp3, line 17: "Observations of NOx canopy fluxes and atmospheric models…"; here you suggest that model studies show that trees take up NOx mixing ratios over 0.1 ppbv. What atmospheric models are those?? I think that models generally produce a whole range of results on NOx fluxes dependent on how the biogenic emissions, dry deposition (and canopy interactions) have been implemented and on the assumptions being made but which up to now lack actually lots of experimental information on issues such as the existence of the compensation point. Here we really need to connect leaf-to canopy-scale and in-canopy NOx gradient and flux measurements together with multilayer exchange models to further demonstrate the potential existence and relevance of leaf- to canopy-scale NOx compensation points for difference ecosystems.*

What was meant in stating, "…trees are substantial sinks…" is that there is still dry deposition of $NO_2$ at even low $NO_x$ mixing ratios, not that the overall canopy flux is negative. The biogenic emissions, dry deposition, and canopy interactions could still, of course, make the forest system a net source of $NO_x$. The line was edited to reflect this. A few sentences were also added to address modelling studies (eg Seok et al., 2013) that have suggested the existence of a compensation point: "Emission of NO at these low NOx mixing ratios has also been detected in laboratory chamber studies (Wildt et al., 1997; Hereid and Monson, 2001). More recent laboratory studies of leaf level deposition have, however, questioned the existence of a compensation point (Chaparro-Suarez et al., 2011; Breuninger et al., 2013). Most observations of NOx canopy fluxes and atmospheric models predict or assume substantial NOx deposition at concentrations as low as 0.1 ppb, typical of NOx mixing ratios in remote areas (Jacob and Wofsy, 1990; Wang and Leuning, 1998; Lerdau et al., 2000; Sparks et al., 2001; Wolfe and Thornton, 2011; Min et al., 2012; Geddes and Murphy, 2014). However, some modelling studies have suggested that a NO2 compensation point is necessary to describe (Seok et al., 2013), or has only a small effect on canopy fluxes in most regions (Ganzeveld et al., 2002a). More research is thus needed on leaf and canopy-level processes to understand the full complexity of the soil-canopy-atmosphere system."

*C4) Page 5, line 19: "This corresponded to a maximum loss of 0.4 ppb at 8 ppb NO2". Can you assume that the wall loss scales linearly with the concentration? What are the wall losses for the minimum concentrations you used for the experiments?*

The wall loss had been directly measured and has been found to scale linearly and has already been subtracted from reported fluxes. The minimum concentration used was 1ppb, corresponding to a wall loss of 0.05 ppb. We revised the manuscript to also state this minimum loss.

*C5) Page 8-9: "For all light and dark experiments the average compensation point for NO was calculated as 0.84 ± 0.32 ppb NO and 2.4 ± 1.1 ppb NO, respectively (Table 2). Page 9: "making Quercus agrifolia a large net sink of NOx"; I see here your point that this tree species seems to be a sink of NOx given that the NO emission flux is only half the NO2 deposition flux but this doesn't confirm so much that this tree species is overall providing a large sink of NOx (which would depend on the overall functioning of the canopy–soil system). Reading over then also later on Section 4.2, this is indeed confirmed having an overall loss by deposition to these trees on the order of 3-7% of total NOx loss in the boundary layer.*

We revised the manuscript page 9 so that the sentence the reviewer referred to specifies that the deposition process, specifically, acts as a sink, not necessarily the canopy-soil system as a whole: "At typical $NO_2/NO$ ratios and gradients measured in forest canopies, the leaf-level $NO_2$ and NO exchange fluxes measured make dry stomatal deposition to *Quercus agrifolia* a net sink of $NO_x$ within the canopy".

*C6) Regarding the presented study on the implications of the leaf-level measurements of NO2/NO compensation points for canopy-scale NOx exchange, there is a study by Seok et al. (Dynamics of nitrogen oxides and ozone above and within a mixed hardwood forest in norther Michigan ACP, 2013) that addressed the potentially important role of the compensation point based on analysis of in and above-canopy NOx concentration dynamics also using a multi-layer model on this dataset. The observed early morning peak of NOx was best explained actually considering the role of a NOx compensation point in the exchange simulations.*

We agree with the reviewer that the study by Seok et al. is quite relevant to the implications for the compensation point discussed in this paper. Seok et al. 2013 was added as a reference in edits made while addressing C3.

We also made the following additional edits to the manuscript:

1)  Errors were found in the calculation of the $NO_2$ cuticular resistance, which was corrected in the text and Table 3. The cuticular resistance is corrected to $65 \pm 8$ s cm$^{-1}$.

2)  Statements were added to address possible errors in the determination of the boundary resistance.

[revised manuscript text omitted]